# Temporospatial Nestedness in Consciousness: An Updated Perspective on the Temporospatial Theory of Consciousness

**DOI:** 10.3390/e25071074

**Published:** 2023-07-17

**Authors:** Zirui Huang

**Affiliations:** 1Department of Anesthesiology, University of Michigan Medical School, Ann Arbor, MI 48109, USA; huangzu@med.umich.edu; 2Center for Consciousness Science, University of Michigan Medical School, Ann Arbor, MI 48109, USA

**Keywords:** neural activity, nestedness, functional geometry, temporal dynamics, consciousness, dimensions, cortical gradient, temporal circuit

## Abstract

Time and space are fundamental elements that permeate the fabric of nature, and their significance in relation to neural activity and consciousness remains a compelling yet unexplored area of research. The Temporospatial Theory of Consciousness (TTC) provides a framework that links time, space, neural activity, and consciousness, shedding light on the intricate relationships among these dimensions. In this review, I revisit the fundamental concepts and mechanisms proposed by the TTC, with a particular focus on the central concept of temporospatial nestedness. I propose an extension of temporospatial nestedness by incorporating the nested relationship between the temporal circuit and functional geometry of the brain. To further unravel the complexities of temporospatial nestedness, future research directions should emphasize the characterization of functional geometry and the temporal circuit across multiple spatial and temporal scales. Investigating the links between these scales will yield a more comprehensive understanding of how spatial organization and temporal dynamics contribute to conscious states. This integrative approach holds the potential to uncover novel insights into the neural basis of consciousness and reshape our understanding of the world–brain dynamic.

## 1. Time and Space in the Brain

Consciousness, a multifaceted and enigmatic phenomenon, has been the subject of exploration from various perspectives. Unraveling the neurobiological underpinnings of consciousness holds great significance for multiple medical disciplines such as neurology, psychiatry, and anesthesiology. The field has witnessed substantial advancements in the past two decades, thanks to various theoretical breakthroughs that have propelled our understanding forward [1,2,3,4,5,6,7,8,9,10,11,12,13,14,15,16].

Time and space serve as fundamental elements in the fabric of nature. Similarly, within the realm of the brain’s neural activity, these elements manifest and propagate in their intrinsic forms [17]. Despite extensive investigations of time and space in the field of physics, their significance in relation to neural activity and consciousness remains largely unexplored. Drawing upon various lines of empirical evidence, Dr. Northoff and I have formulated the Temporospatial Theory of Consciousness (TTC) [18]. By elucidating the intricate relationships among time, space, neural activity, and consciousness, the TTC offers a framework that links various types of neural activity with different aspects of consciousness [18,19,20]. In this article, I will revisit the fundamental concepts and mechanisms proposed by the TTC and expand on its central concept of temporospatial nestedness. To achieve this, I will update the definition of consciousness from a multidimensional perspective and incorporate recent research on the spatial rules of brain activity origination and the temporal regularities of brain activity.

When we refer to space and time in the brain, we are talking about the extension of neural activity across different hierarchical spatial scales, ranging from neuronal layers and regions to functional networks, as well as the duration of neural activity embedded in hierarchical timescales [17]. In the visual system, for example, there are fine-grained projections from the retina to the cortex, and different neurons have their own preference for object orientation [21,22,23]. This information is integrated through different spatial scales. Likewise, in the context of receiving sensory stimuli across time, such as listening to a story, our brain processes the information by employing distinct temporal receptive windows [24]. For instance, words are processed within the span of hundreds of milliseconds, sentences within a few seconds, and paragraphs within tens of seconds [25,26]. Brain regions characterized by higher-frequency fluctuations play a role in processing swift sensory information, whereas regions exhibiting dominance in low frequencies are involved in integrating perceptual and cognitive events that unfold over longer durations [24,25,26,27]. By exploring these temporal and spatial aspects of the brain, the TTC provides a unique perspective on the nature of consciousness and its neural underpinnings. Four mechanisms were proposed to account for different aspects of consciousness involving the temporospatial nestedness in spontaneous brain activity, the temporospatial alignment during the interactions between pre- and post-stimulus-induced activity, the temporospatial expansion of early stimulus-induced activity, and the temporospatial globalization of late stimulus-induced activity.

## 2. Four Mechanisms in the TTC

Mechanism 1, temporospatial nestedness (Figure 1), refers to the systematic cross-frequency coupling [28,29,30,31,32,33,34,35,36,37,38] and hierarchical intrinsic timescales in the brain’s temporal domain [20,27,39,40,41,42,43,44,45,46,47,48,49,50,51,52,53,54,55], as well as the hierarchical organization across brain regions in the spatial domain [49,56,57,58,59,60]. These dynamic, complex, and flexible temporospatial configurations are crucial in generating our rich phenomenal experiences [17]. Certain changes in the cross-frequency relationship [61,62,63,64], intrinsic timescales [65,66], or functional network hierarchy [67,68] may lead to a loss of consciousness.

The first instance of Mechanism 2, temporospatial alignment (Figure 2), is single-stimulus alignment. The concept behind this is that a stimulus needs to be presented at the appropriate moment for it to interact with the spontaneous activity of the brain, enabling us to perceive it. For example, if sub-threshold sounds are presented, sometimes we can hear them, and sometimes we cannot. This is because the spontaneous activity has an impact on our perception. If the stimulus arrives during the preferred phases of spontaneous activity, we are able to detect it, but not during other phases [69,70,71,72,73,74,75,76,77,78,79,80,81,82]. Another example is sequential temporal alignment in spatially distributed functional areas (see also a three-layer temporal model of alignment in [83]), which we previously discussed regarding temporal receptive windows. A third example is long-term alignment across the lifespan, and the key takeaway is that the alignment between environment-related signals and organism-related signals enables the organism to model the relationship between itself and the world [16,18,83]. The Integrated World Modeling Theory (IWMT) [84,85] has provided a sophisticated elaboration of this notion, proposing that consciousness arises from generative processes that integrate information into coherent models encompassing space, time, and causal relationships between systems and their environments. The IWMT shows promise in providing a mechanistic understanding of temporospatial alignment through its thorough integration with computational models.

Mechanism 3, temporospatial expansion (Figure 3), proposes that the brain’s neural activity needs to expand both in time and space for conscious experience to occur. The idea is that when we are presented with a sensory stimulus, such as a picture of a face, the neural activity initially activates low-level sensory areas, like the primary visual cortex, but we are not yet able to perceive anything. As the activity expands in time and space to a certain level, we begin to have conscious experience or qualia. Studies using techniques like transcranial magnetic stimulation (TMS) and electroencephalography (EEG) support the notion that the conscious brain propagates the stimulus-evoked activity both in time and space [86], while the unconscious brain may lose either the spatial expansion or temporal expansion or both. Another example of temporospatial expansion is seen in the fact that the duration and extension of a stimulus in our conscious experience usually last longer and extend beyond its physical duration and extension [87]. For instance, when we watch a movie, there are temporal gaps between frames and spatial gaps between pixels, yet we still have a unified experience. The temporospatial expansion mechanism may provide a “glue” between the stimulus and the brain, yielding the continuity of our phenomenal experience. The idea of temporospatial expansion is also compatible with the posterior hot zone hypothesis [6,88], which suggests that when the expansion passes a threshold both in time and space in the posterior part of our brain, we become capable of having phenomenal experiences.

Mechanism 4, temporospatial globalization (Figure 4), is the process by which the temporal and spatial aspects of neural activity become globally available for report, resulting in access consciousness [3,89,90]. In other words, once the temporospatial expansion is globalized, involving the lateral frontoparietal loops, conscious content can be accessed and reported. This mechanism is predicted by the Global Neuronal Workspace Theory [15,91], which suggests that conscious information is made globally available when it is broadcast to a large network of neurons in the brain. This broadcasting is thought to be facilitated by neural signatures such as the P3b and non-linear ignition [91,92,93]. The concept of temporospatial globalization complements the Global Neuronal Workspace Theory by adding the dimension of temporal globalization on top of cognitive and neuronal space.

To this end, I have reviewed the four mechanisms of consciousness proposed by TTC, and I argue that Mechanism 1 serves as the foundational building blocks of consciousness that enable the other mechanisms to operate (for a similar view, see [17]). Moving forward, the focus will be on expanding upon the central concept of the TTC, temporospatial nestedness, which is key to understanding the state of consciousness. This will involve updating the definition of consciousness from a multidimensional perspective and incorporating recent studies on the spatial organization and temporal regularities of brain activity. The goal is to gain a deeper understanding of the relationship between consciousness and the brain’s temporospatial patterns, and how alterations in these patterns relate to changes in consciousness.

## 3. Dimensions of Consciousness

For the past two decades, scientists have emphasized the need to identify key “dimensions” of consciousness. In the field of anesthesiology, consciousness is typically viewed as a variable that changes along a single dimension that is often equated with the level of arousal [95]. In neurology, consciousness is typically viewed as consisting of two dimensions: (1) awareness of the environment and self (also known as the content of consciousness); and (2) wakefulness (also known as the level of consciousness) [96]. While these two dimensions typically coexist, they can become dissociated in everyday situations (e.g., sleep) or under the influence of pharmacological or pathological conditions [97]. For example, patients with unresponsive wakefulness syndrome (vegetative state) have their eyes wide open but are presumably unaware of themselves and their surroundings. Therefore, their condition is considered wakefulness without awareness. This awareness versus wakefulness scheme was later expanded by adding a behavioral dimension (i.e., the ability to produce motor response) [98], which allows for a cognitive–motor dissociation [99] or covert consciousness [100] that occurs in some neuropathological patients who are behaviorally non-responsive. Although these approaches to defining consciousness along multiple cognitive–behavioral dimensions are useful for the clinical diagnosis of neuropathological patients, they controversially assume that normal conscious wakefulness is associated with the highest level of awareness, wakefulness, and responsiveness. Altered states of consciousness, such as those induced by psychedelic drugs or psychiatric causes (e.g., schizophrenia), are difficult to fit into such a scheme, as these patients can be fully aware, awake, and responsive but with even richer conscious experiences than in a drug-free state [101].

In recent years, additional dimensions of consciousness have been proposed, including cognitive dimensions like attention control, sensory processing, executive function, and meta-awareness [102,103,104,105,106]. Despite their theoretical appeal, these dimensions have yet to be precisely linked to neural activity and function in the brain. Although research has extensively examined the neural activity associated with wakefulness and awareness, it has not been performed within the context of mapping a single conceptual dimension of consciousness to a measurable neural variable. One factor contributing to this lack of understanding is the failure of traditional neuroimaging localization approaches to connect specific brain regions with exclusive functions. For instance, the prefrontal cortex serves several functions, such as working memory, decision making, attention, and task control [107]. Additionally, a single brain network can be implicated in multiple cognitive processes at the system level [108]. Therefore, brain regions and networks do not correspond to unique neurofunctional dimensions. In the quest for the neural correlates of consciousness [6], researchers have generally examined distinct brain regions instead of continua reflecting the brain’s intrinsic functional geometry.

## 4. Functional Geometry—Spatial Organization Rules

Recent advances in neuroimaging offer a new approach that can be applied to investigate the neural dimensions of consciousness by representing the brain’s functional geometry in terms of its cortical gradients [56,109]. These gradients encompass various functions and networks along a continuum that ranges from perception and action to abstract cognitive functions [56,110,111,112]. This functional spectrum is represented by the unimodal to transmodal gradient (Gradient-1), which facilitates the integration of sensory signals with non-sensory information and transforms them into abstract contents. The visual to somatomotor gradient (Gradient-2) represents the functional specialization of different sensory modalities. The task-negative to task-positive gradient (Gradient-3) represents a functional differentiation extending from areas that are often deactivated during task performance to those activated during tasks in frontoparietal and attention networks [113,114]. The cortical gradient mapping approach allows for a more comprehensive understanding of the neural dimensions of consciousness by assessing the role of the topographical continuum, which is analogous to describing a region along several topographical axes related to the slope of elevation or spatial change in vegetation types (Figure 5).

Using cortical gradient mapping, recent research has shown that the three cortical gradients (Gradient-1, Gradient-2, and Gradient-3) may represent the neurofunctional dimensions of consciousness, such as awareness, sensory organization, and cortical arousability [68]. The study found that these gradients selectively changed in various pharmacologically altered states [68]. For example, the degradation of Gradient-1 was observed in deep sedation or general anesthesia induced by propofol, indicating a loss of awareness. However, the degradation of Gradient-1 was less pronounced during ketamine anesthesia, suggesting partial preservation of inner awareness, such as dream-like experiences. In contrast, ketamine anesthesia resulted in a significant degradation of Gradient-2, indicating increased visual-somatosensory crosstalk, which is a common sensory disorganization effect of psychoactive drugs. Finally, Gradient-3 degraded when subjects were no longer arousable by painful glabellum stimulation during propofol infusion at a dose sufficient for general anesthesia. Taken together, these findings provide a brain-based understanding of consciousness as a multidimensional phenomenon (Figure 6) and offer a theoretical and empirical framework for unraveling the complex brain mechanisms underlying consciousness.

## 5. Temporal Circuit—Regularities in Time

The brain is in a constant state of flux, continually altering its functional connections and evolving over time. In a recent study, researchers employed an unsupervised machine learning technique called k-means clustering to examine moment-to-moment brain activity, identifying eight distinct co-activation patterns (CAPs) [115]. Two of these CAPs correspond to the default mode and dorsal attention networks, while the remaining six are associated with other well-known networks involved in various brain functions, including the frontoparietal, sensory and motor, visual, and ventral attention networks, as well as two networks representing cross-brain states of activation and deactivation. Interestingly, these eight CAPs form four pairs of “mirror” motifs, with a strong negative spatial similarity between each pair (Figure 7).

Furthermore, the work reveals how the push–pull relationship between the default mode and dorsal attention networks may differ between conscious and unconscious individuals [115]. In the conscious brain, the dynamic switching of networks including the default mode and dorsal attention networks occurs along a set of structured transition trajectories, which can be considered a “temporal circuit”. Disruption of this circuit, resulting in limited access to these networks, is a common characteristic of unconsciousness resulting from various causes. In a subsequent study [116], it was found that the ventral attention network plays a crucial role in modulating the transitions between the default mode and dorsal attention networks, which aligns with the triple network hypothesis [117,118]. Furthermore, the ventral attention network operates at short timescales [49,119], facilitating rapid detection and orientation toward salient stimuli [120]. The dorsal attention network, on the other hand, operates at medium timescales [49,119], enabling sustained attention and cognitive control [120,121]. Lastly, the default mode network exhibits temporal dynamics in long timescales that are involved in long-term memory, imagination, and self-referential processes [49,119,122,123]. By considering these nested temporal scales of the network interactions within the triple network hypothesis, we gain a better understanding of the distinct temporal characteristics and functions of each network.

Why are the default mode and dorsal attention networks important to consciousness? They play a crucial role as they correspond to our inward focus on ourselves and outward focus on the environment, respectively [120,121,123,124,125,126,127]. In the conscious brain, these two systems are in a dynamic balance, sliding back and forth, but both are present to some extent. The temporal circuit hypothesis posits that consciousness emerges from the reciprocal balance between these two opposing cortical systems, embedded in the spatiotemporal dynamics of neural activity. To be conscious, an agent must accurately represent its environment and also represent itself in relation to aspects of the world, either external or internal. This is why the self is crucial in the emergence of consciousness, as it is the subject that is aware of the environment, and without a subject that is aware, there is no consciousness [84,85,128,129,130,131]. The dynamic relation between the self and the environment, mediated by the default mode and dorsal attention networks, may enable consciousness to emerge.

## 6. Temporal Circuit Nested within Functional Geometry—Temporospatial Nestedness

Recent research suggests that the spatial and temporal properties of brain activity are interdependent and cannot be fully understood in isolation. Studies show that transient fMRI co-activations propagate as waves along cortical gradients [132,133,134,135], indicating that temporal dynamics are likely constrained by cortical gradients. Therefore, understanding the covariation between spatial and temporal dimensions can provide greater insight into the neural basis of consciousness. The aforementioned three cortical gradients represent major neurofunctional dimensions of consciousness, and together create a three-dimensional gradient space. Functional brain networks occupy distinct positions in this space, and their functional geometry, such as distance, can be quantified. The network’s functional geometry and the temporal occurrence rates of corresponding co-activation patterns were found to covary with the state of consciousness [68]. For example, the degradation of functional gradients is linked to the disruption of the temporal circuit in depressed states of consciousness (Figure 8). These findings provide insight into the interplay between the brain’s functional geometry and its temporal dynamics, highlighting their potential nested relationship. This notion of nestedness, which is a core concept in the TTC, can be extended by including both spatial (i.e., functional geometry) and temporal (e.g., temporal circuit) dimensions of consciousness.

## 7. Concluding Remarks

The TTC offers a unique perspective on the nature of consciousness and its neural correlates by exploring the temporal and spatial dimensions of the brain’s neural activity. With its framework, the TTC provides insights into various aspects of consciousness. One of its key concepts, temporospatial nestedness, can be extended by including a nested relationship between the functional geometry and the temporal circuit of the brain. To gain a better understanding of this relationship, future research should focus on characterizing functional geometry and the temporal circuit across multiple spatial and temporal scales, as well as investigating their links across scales. By doing so, we can achieve a more comprehensive understanding of temporospatial nestedness and possibly uncover new insights into the neural basis of consciousness. Together, the TTC opens up new avenues of research, paving the way for innovative investigations into the interplay between the brain and consciousness and fostering a deeper appreciation for the world–brain dynamic.

## Figures and Tables

**Figure 1 entropy-25-01074-f001:**
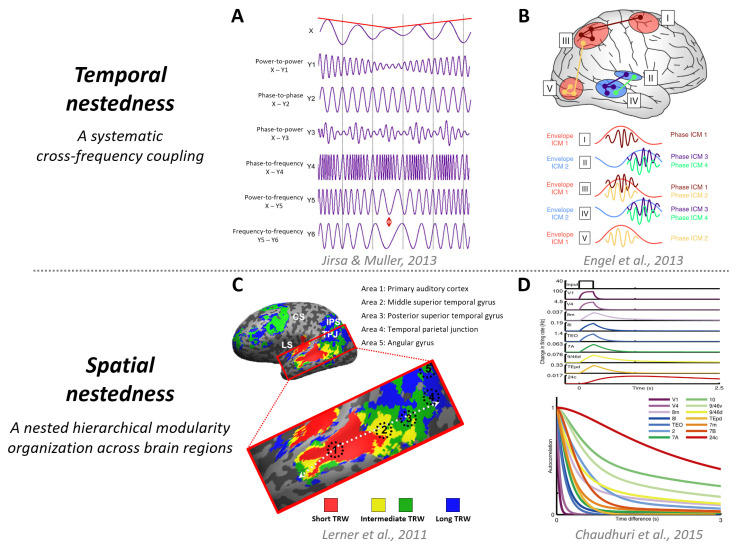
Temporospatial nestedness. (**A**) Various types of cross-frequency coupling, including power-to-power, phase-to-phase, phase-to-power, phase-to-frequency, power-to-frequency, and frequency-to-frequency. The depiction is adapted from the work of Jirsa and Müller (2013) [28]. (**B**) Interaction between envelope and phase for intrinsic coupling modes (ICMs). Envelope ICMs may serve to regulate the coactivation of brain regions participating in a functional network. The illustration is adapted from the work of Engel et al. (2013) [38]. (**C**) Hierarchical topography of temporal receptive windows (TRWs) mapped using functional magnetic resonance imaging (fMRI) in response to audio narratives. The fMRI map demonstrates the gradual transition from short to long TRWs along the temporal–parietal axis. The illustration is adapted from the work of Lerner et al. (2011) [26]. (**D**) Hierarchy of timescales in response to visual input. A pulse of input to area V1 is propagated along the hierarchy, displaying increasing decay times as it progresses. The autocorrelation of area activity in response to white noise input to V1 shows a functional hierarchy ranging from area V1 at the bottom to prefrontal areas at the top. The illustration is adapted from the work of Chaudhuri et al. (2015) [39].

**Figure 2 entropy-25-01074-f002:**
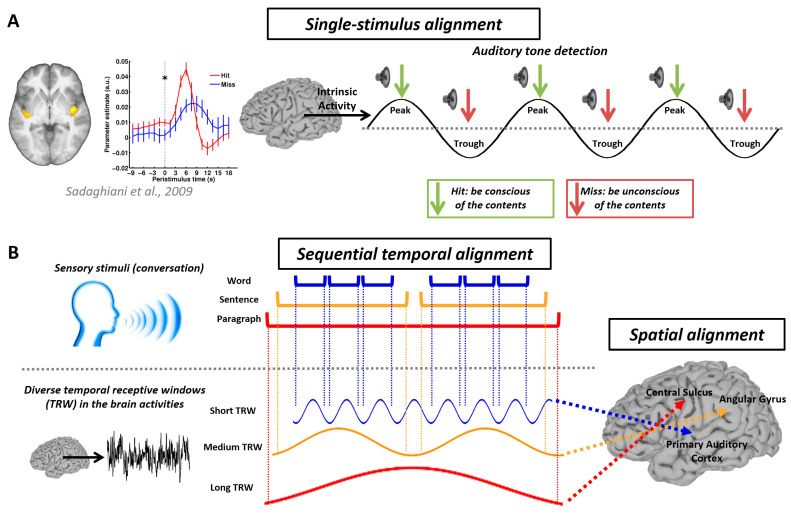
Temporospatial alignment. (**A**) Pre-stimulus functional magnetic resonance imaging (fMRI) time courses from the bilateral auditory cortex. It shows significantly (as indicated by *) higher activity preceding hits compared to misses at the stimulus onset (t = 0; as indicated by a dash line). The illustration is adapted from the work of Sadaghiani et al. (2009) [72]. (**B**) How the brain processes information by employing distinct temporal receptive windows (TRWs). Words are processed within the span of hundreds of milliseconds, sentences within a few seconds, and paragraphs within tens of seconds. Brain regions characterized by higher frequency fluctuations contribute to the processing of swift sensory information, while regions exhibiting dominance in low frequencies are involved in integrating perceptual and cognitive events that unfold over longer durations [24,25,26,27].

**Figure 3 entropy-25-01074-f003:**
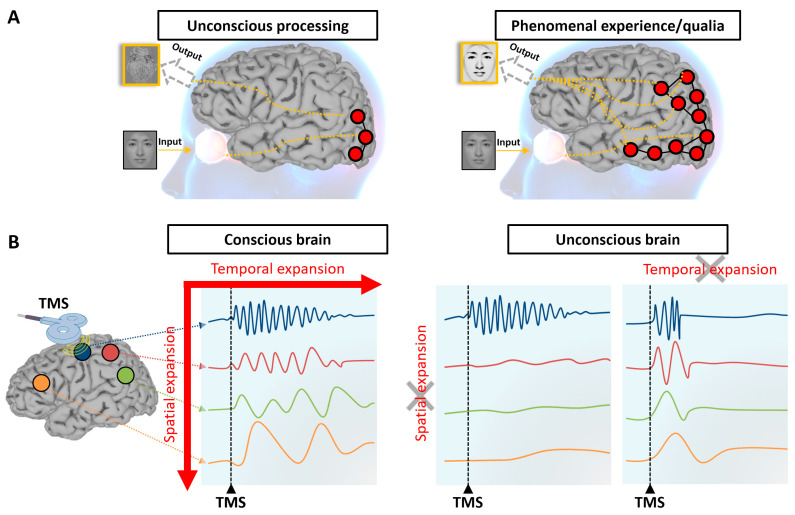
Temporospatial expansion. (**A**) The content of external inputs, such as a face, remains unconscious during low-level sensory registration, with limited temporospatial expansion in the brain. Subjective and phenomenal experience occurs when the temporospatial expansion in brain activity reaches a certain extent, which is associated with the state of “no-report” consciousness. (**B**) A conscious brain can propagate the TMS -evoked activity both in time and space. In contrast, an unconscious brain may lose either the spatial expansion or temporal expansion or both. The illustration is inspired by the works of Casali et al. (2013) [86] and Koch et al. (2016) [6].

**Figure 4 entropy-25-01074-f004:**
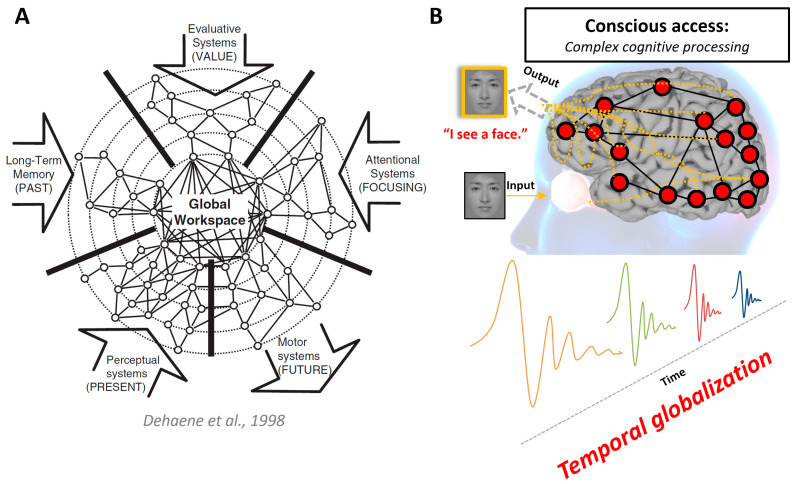
Temporospatial globalization. (**A**) The original depiction of the Dehaene–Changeux model of a Global Neuronal Workspace (GNW) (adapted from [93,94]). The GNW model suggests that subjective experience corresponds to the global availability of information. According to this model, conscious access occurs when a piece of information enters a distributed network of cortical areas that are tightly interconnected by long-distance axons, known as the GNW. This network enables the flexible broadcasting of information to various specialized processors within the brain. (**B**) Subjects are able to “report” the content of external inputs when the temporospatial expansion is globalized. This globalized expansion involves the engagement of lateral frontoparietal networks. The phenomenon of “temporospatial globalization” is concurrent with complex cognitive processing.

**Figure 5 entropy-25-01074-f005:**
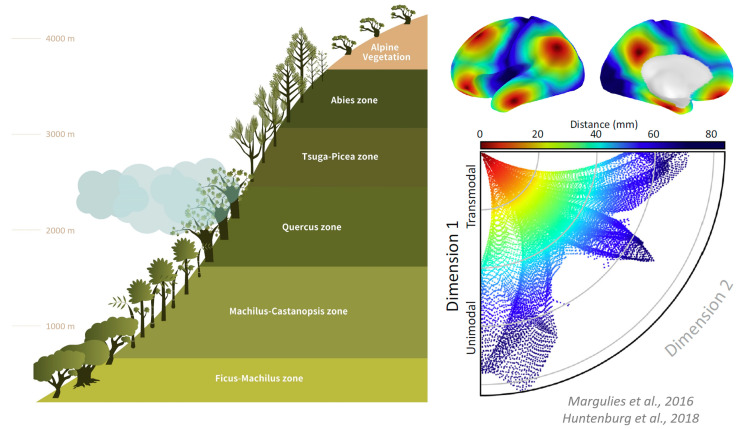
Analogy between land spatial geometry and cortical gradients. Huntenburg et al. (2018) [110] proposed an intrinsic coordinate system based on the distance along the cortex. The dimension from unimodal to transmodal processing is represented by the distance along the cortical surface. The illustration is adapted from the works of Margulies et al. (2016) and Huntenburg et al. (2018) [56,110].

**Figure 6 entropy-25-01074-f006:**
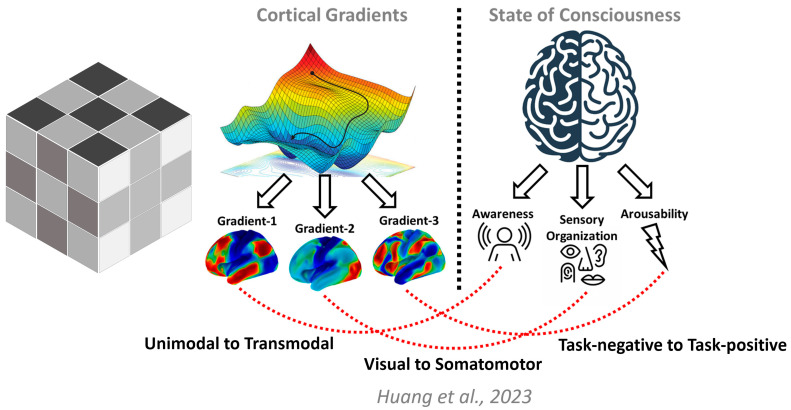
Links between cortical gradients and neurofunctional dimensions of consciousness. Consciousness is often compared to solving a scrambled Rubik’s cube, where examining a single surface may lead to confusion about its organization. To fully understand consciousness, it is necessary to consider multiple dimensions. These dimensions encompass (1) awareness, or what we actually experience, like the redness of a rose; (2) sensory organization, or how sights and sounds and feelings become woven together to create our seamless conscious experience; and (3) arousability, that is, the ability of the brain to be awake. Recent research has identified three cortical gradients that appear to align with these dimensions of consciousness. The illustration is adapted from the work of Huang et al. (2023) [68].

**Figure 7 entropy-25-01074-f007:**
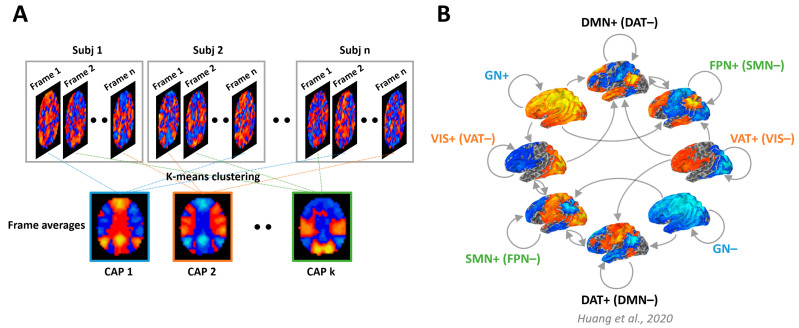
Dynamic brain states and the temporal circuit. (**A**) Illustration of the k-means clustering algorithm used to partition fMRI image volumes into k co-activation patterns (CAPs). (**B**) Eight CAPs were classified as follows: default mode network (DMN+), dorsal attention network (DAT+), frontoparietal network (FPN+), sensory and motor network (SMN+), visual network (VIS+), ventral attention network (VAT+), and global network of activation and deactivation (GN+ and GN−). These eight CAPs can be grouped into four pairs of “mirror” motifs (represented by four color classes). For example, the DMN+ is accompanied by co-deactivation of DAT (DAT−) and vice versa for DAT+ (DMN−). The illustration is adapted from the work of Huang et al. (2020) [115].

**Figure 8 entropy-25-01074-f008:**
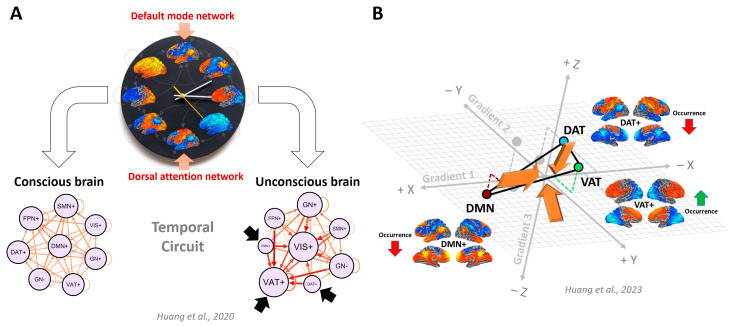
Covariation of the temporal circuit and functional geometry with changes in conscious state. (**A**) The temporal circuit is characterized by trajectories along which dynamic brain states occur. Transitions between the default mode network (DMN+) and dorsal attention network (DAT+) are embedded in this temporal circuit, with balanced reciprocal accessibility of brain states being characteristic of consciousness. Unconsciousness is associated with isolation (i.e., reduced accessibility) of DMN+ and DAT+ from the trajectory space, which is dominated by a few giant attractors such as the visual network (VIS+) and ventral attention network (VAT+). (**B**) The brain’s functional geometry, such as network distance in the gradient space, and temporal dynamics, such as occurrence rates of co-activation patterns, covary with the state of consciousness. Co-activation patterns of DMN+, DAT+, and VAT+ can be considered snapshots of global propagation waves. During unconscious states, the propagation of co-activations across the DMN and DAT (DMN+ and DAT+) becomes less frequent (e.g., low occurrence rates), while those involving the VAT (VAT+) become more frequent. These effects are associated with a reduced functional distance between DMN and VAT, as well as a reduced functional distance between DMN and DAT, in the 3D gradient space. The illustration is adapted from the works of Huang et al. (2020, 2023) [68,115].

## Data Availability

No new data were created.

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
