# Peer review of "Temporospatial Nestedness in Consciousness: An Updated Perspective on the Temporospatial Theory of Consciousness"

_entropy, 2023, doi:10.3390/e25071074_

Round 1

Reviewer 1 Report

Thanks for the opportunity to review this paper on the TTC. It focuses on the notion of temporospatial nestedness in preference to the other three aspects of the TTC - alignment, expansion, and globalisation. I can see the logic for concentrating on nestedness as the primary motif in TTC and think this is a valuable contribution to the literature. The other mechanisms are implicit in nestedness in that to be nested requires alignment of temporal and spatial scales, expansion from one scale to another, and globalisation of particular scales for complex processing. 

I only have a few minor points and one request for a deletion:

Please provide a reference for the bit about temporal windows for syllables, words, and sentences.

Please change the word continuums to continua

And please delete the section on the Yin Yang analogy. I don't see the value in this analogy here other than that there are 8 categories in both instances. Unless you are going to go deep into the analogy, it seems like an unnecessary detour. This would need to be a whole paper in a more suitable journal on metaphysics.

With regard to the triple-network hypothesis, I would like to see something more concrete here regarding the triple-network hypothesis that includes the nested temporal scales of all three networks. Salience network = fast timescales (milliseconds), dorsal attention = medium timescales (working memory 30s), and default mode = single seconds and long-term memory.

Author Response

Reviewer #1

Thanks for the opportunity to review this paper on the TTC. It focuses on the notion of temporospatial nestedness in preference to the other three aspects of the TTC - alignment, expansion, and globalisation. I can see the logic for concentrating on nestedness as the primary motif in TTC and think this is a valuable contribution to the literature. The other mechanisms are implicit in nestedness in that to be nested requires alignment of temporal and spatial scales, expansion from one scale to another, and globalisation of particular scales for complex processing.

Response: I would like to express my gratitude to the reviewer for providing the positive feedback on my manuscript. I appreciate the reviewer's thoughtful comments and suggestions, and I am pleased to address each concern in detail. Changes made in the manuscript are highlighted in red font.

I only have a few minor points and one request for a deletion:

Please provide a reference for the bit about temporal windows for syllables, words, and sentences.

Response: Thank you for your suggestion. I have revised the statement accordingly, incorporating relevant references.

“Likewise, in the context of receiving sensory stimuli across time, such as listening to a story, our brain processes the information by employing distinct temporal receptive windows [24]. For instance, words are processed within the span of hundreds of milliseconds, sentences within a few seconds, and paragraphs within tens of seconds [25,26].”

  1. Hasson, U.; Yang, E.; Vallines, I.; Heeger, D.J.; Rubin, N. A Hierarchy of Temporal Receptive Windows in Human Cortex. J Neurosci 2008, 28, 2539–2550, doi:10.1523/jneurosci.5487-07.2008.
  2. Hasson, U.; Chen, J.; Honey, C.J. Hierarchical Process Memory: Memory as an Integral Component of Information Processing. Trends Cogn Sci 2015, 19, 304–313, doi:10.1016/j.tics.2015.04.006.
  3. Lerner, Y.; Honey, C.J.; Silbert, L.J.; Hasson, U. Topographic Mapping of a Hierarchy of Temporal Receptive Windows Using a Narrated Story. J Neurosci 2011, 31, 2906–2915, doi:10.1523/jneurosci.3684-10.2011.

Please change the word continuums to continua

Response: Thank you for bringing this to my attention. I have changed the word "continuums" to "continua."

And please delete the section on the Yin Yang analogy. I don't see the value in this analogy here other than that there are 8 categories in both instances. Unless you are going to go deep into the analogy, it seems like an unnecessary detour. This would need to be a whole paper in a more suitable journal on metaphysics.

Response: I fully agree that the Yin Yang analogy may not significantly contribute to the main focus of the paper. Therefore, I have followed the recommendation and deleted the section on the Yin Yang analogy. Furthermore, I have provided an updated Figure 7 to reflect this change. I believe that these revisions will enhance the clarity and focus of the paper.

With regard to the triple-network hypothesis, I would like to see something more concrete here regarding the triple-network hypothesis that includes the nested temporal scales of all three networks. Salience network = fast timescales (milliseconds), dorsal attention = medium timescales (working memory 30s), and default mode = single seconds and long-term memory.

Response: Thanks for the suggestion regarding the triple-network hypothesis and the inclusion of nested temporal scales for all three networks. I agree that providing more concrete information in this regard would strengthen the discussion. I have revised the relevant paragraph as follows:

“In a subsequent study [116], it was found that the ventral attention network plays a crucial role in modulating the transitions between the default-mode and dorsal attention networks, which aligns with the triple-network hypothesis [117,118]. Furthermore, the ventral attention network operates at short timescales [49,119], facilitating rapid detection and orientation towards salient stimuli [120]. The dorsal attention network, on the other hand, operates at medium timescales [49,119], enabling sustained attention and cognitive control [120,121]. Lastly, the default mode network exhibits temporal dynamics in long timescales that are involved in long-term memory, imagination, and self-referential processes [49,119,122,123]. By considering these nested temporal scales of the network interactions on the triple-network hypothesis, we gain a better understanding of the distinct temporal characteristics and functions of each network.”

Reviewer 2 Report

The aim of this paper was to provide an overview of the fundamental concepts and mechanisms proposed by the Temporospatial Theory of Consciousness (TTC), with a particular focus on the central concept of temporospatial nestedness. The author further proposed an extension of temporospatial nestedness by incorporating the nested relationship between the functional geometry and temporal circuit of the brain. This is novel and interesting development.

I have found the paper very important for the field of consciousness studies. Especially, the discussion of the cortical gradients that allow for a more comprehensive understanding of the neural dimensions of consciousness by assessing the role of the topographical continuum, is very important and novel. Additionally, I find the usage of analogy between the neural motifs and the yin-yang Bagua theory quite interesting and intriguing. However, I have a feeling that some important previous publications are not mentioned in this manuscript and it makes this manuscript less connected to (and disconnected from) a previous work. I also have a few small other suggestions and comments. I recommend to accept this manuscript for publication after a minor revision.

Below I present several comments/suggestions, that I believe may improve the manuscript. The manuscript does not have pagination, so it is not easy to point to a precise location in the text, while commenting.

1. Section 'Time and space in the brain'; 2nd paragraph; the sentences: "Time and space serve as fundamental elements in the fabric of nature. Similarly, within the realm of the brain's neural activity, these elements manifest and propagate in their intrinsic forms." One of the first extensive discussions and conceptualizations on this topic was provided in the 2010 paper: "Natural world physical, brain operational, and mind phenomenal space–time" doi:10.1016/j.plrev.2010.04.001 . For example, this paper states: "Concepts of space and time are widely developed in physics. However, there is a considerable lack of biologically plausible theoretical frameworks that can demonstrate how space and time dimensions are implemented in the activity of the most complex life-system—the brain with a mind. Brain activity is organized both temporally and spatially, thus representing space–time in the brain. [...] At the same time, to have a fully functional human brain one needs to have a subjective mental experience. Current research on the subjective mental experience offers detailed analysis of space–time organization of the mind. According to this research, subjective mental experience (subjective virtual world) has definitive spatial and temporal properties similar to many physical phenomena. Based on systematic review of the propositions and tenets of brain and mind space–time descriptions, our aim in this review essay is to explore the relations between the two. To be precise, we would like to discuss the hypothesis that via the brain operational space–time the mind subjective space–time is connected to otherwise distant physical space–time reality." It is important to include the reference to this paper in relation to the mentioned sentences in the revised manuscript .

2. Section 'Time and space in the brain'; 3nd paragraph; the sentence: "When we refer to space and time in the brain, we are talking about the extension of neural activity across different hierarchical spatial scales, ranging from neuronal layers and regions to functional networks, as well as the duration of neural activity embedded in the hierarchical timescales." Again, the reference to the previously mentioned paper ("Natural world physical, brain operational, and mind phenomenal space–time" doi:10.1016/j.plrev.2010.04.001) is important here.

3. Section 'Four mechanisms in TTC'. 1st paragraph; the sentence: "The dynamic, complex, and flexible temporospatial configurations are crucial in generating our rich phenomenal experiences." Again the reference to a previously suggested paper ("Natural world physical, brain operational, and mind phenomenal space–time" doi:10.1016/j.plrev.2010.04.001) is important here.

4. Section 'Four mechanisms in TTC'. 1st paragraph; the sentence: "Changes in the cross-frequency relationship [60–63], intrinsic timescales [64,65], or functional network hierarchy [66,67] can lead to a loss of consciousness." I would suggest to start this sentence as: "Certain changes in ...", because not all changes lead to a loss of consciousness. Also, I would suggest to change the "... can lead to ..." to "... may lead to ...".

5. Section 'Four mechanisms in TTC'. 2nd paragraph (after the Figure 1); the sentence: "A third example is long-term alignment across the lifespan, and the key takeaway is that the alignment between environment-related signals and organism-related signals enables the organism to model the relationship between itself and the world." I would suggest that the author provides a proper reference here. 

6. Section 'Four mechanisms in TTC'. 3nd paragraph (after the Figure 2); the sentence: "Another example of temporal-spatial expansion is seen in the fact that the duration and extension of a stimulus in our conscious experience usually last longer and extend beyond its physical duration and extension." A complimentary discussion is provided in the paper: "Present moment, past, and future: mental kaleidoscope" doi: 10.3389/fpsyg.2014.00395 . I would suggest to make a reference to it here.

7. Figure 3. Normally, the Figures should be self-explanatory, meaning that the Figure legend contains all necessary information to understand the Figure without reference to the text. I would suggest to add such legends to all Figures to explain the abbreviations and terms (if any).

8. Section 'Four mechanisms in TTC'. 4th paragraph (after the Figure 3); the sentence: "This mechanism is based on the Global Neuronal Workspace Theory [15,86], which suggests that ....". I would suggest to re-frame the phrase: "This mechanism is based on the Global Neuronal Workspace Theory". The mechanism could not be based on a theory - it could be predicted by the theory or the theory could be based on the known mechanism...

9. Section 'Four mechanisms in TTC'. 5th paragraph (after the Figure 4); the sentence: "To this end, we have reviewed the four mechanisms of consciousness proposed by TTC, and I argue that Mechanism 1 serves as the foundational building blocks of consciousness that enable the other mechanisms to operate." I would suggest to add in the end of this sentence: (for a similar view see, reference to "Natural world physical, brain operational, and mind phenomenal space–time" doi:10.1016/j.plrev.2010.04.001).

10. Section 'Dimensions of consciousness'; 1st paragraph; the sentence: "While these two dimensions usually go together, they can be dissociated under pharmacological or pathological conditions." A reference to a paper that is dedicated to this issue is missing here: "Do we need a theory-based assessment of consciousness in the field of disorders of consciousness?" doi: 10.3389/fnhum.2014.00402 . For example this paper states: "Consciousness is often conceptualized as a phenomenon with two components: wakefulness and awareness (Posner et al., 2007). Though such understanding is currently quite wide-spread, it confuses and mixes two different and independent phenomena: subjective awareness and vigilance. While awareness is an important component of consciousness, wakefulness belongs to the vigilance domain. Independence of these two concepts can be demonstrated by examples from a daily life: (a) we are able to unconsciously perform complex actions like brushing our teeth or driving a car while being completely awake; (b) being at the same level of wakefulness we are usually aware of some vents/stimuli while unaware of others; and (c) during sleep we can be aware of our phenomenal experience (dreams) but are not awake. Hence, wakefulness is not a component of consciousness but of vigilance. Vigilance, however, affects consciousness by limiting the amount of information available for conscious access (Rusalova, 2006), thus affecting the amount of content". I would suggest to make a reference to this paper in the end of the sentence above. Also, as it is evident from the provided citation, the dissociation between awareness and wakefulness is also possible in everyday situations. In this respect, I would suggest to reflect this issue in the sentence above.

11. Section 'Functional geometry - spatial organization rules'; 2nd paragraph (after the Figure 5); the sentence: "The study found that these gradients selectively changed in various pharmacologically altered states."  I would suggest that the author provide a proper reference here.

12. Figure 7. As I have mentioned before, please provide a legend explaining the abbreviations and the meaning of '+' and '-'. Also, some terminology from the Yin-Yang Bagua theory should be explained for those who are not familiar with them: the confusing terms are "small/big earth"; "small/big metal"; "small/big wood". 

13. Section 'Temporal circuit - regularities in time'; 3rd paragraph; the sentence: "This is why the self is crucial in the emergence of consciousness, as it is the subject that is aware of the environment, and without a subject that is aware, there is no consciousness [119,120]." An extensive discussion and the empirical support may be found in the paper "Selfhood triumvirate: From phenomenology to brain activity and back again" https://doi.org/10.1016/j.concog.2020.103031 . I would suggest to add reference to this work here.

14. Figure 8. As proposed before, I would suggest to include the legend to this Figure, explaining the abbreviations, and some of the arrows: black arrows, orange arrows, red and green arrows. Also the word "occurrence" does not tell the reader anything meaningful. Please explain in the legend what does it mean in this context.

15. Acknowledgments. Neither Dr. Hudetz, not Dr Mashour are authors of this manuscript. So it is confusing to mention their names in a way they are mentioned. My suggestion is instead of "(to ...)" write "(principle investigator ...)".

Author Response

Reviewer #2

The aim of this paper was to provide an overview of the fundamental concepts and mechanisms proposed by the Temporospatial Theory of Consciousness (TTC), with a particular focus on the central concept of temporospatial nestedness. The author further proposed an extension of temporospatial nestedness by incorporating the nested relationship between the functional geometry and temporal circuit of the brain. This is novel and interesting development.

I have found the paper very important for the field of consciousness studies. Especially, the discussion of the cortical gradients that allow for a more comprehensive understanding of the neural dimensions of consciousness by assessing the role of the topographical continuum, is very important and novel. Additionally, I find the usage of analogy between the neural motifs and the yin-yang Bagua theory quite interesting and intriguing. However, I have a feeling that some important previous publications are not mentioned in this manuscript and it makes this manuscript less connected to (and disconnected from) a previous work. I also have a few small other suggestions and comments. I recommend to accept this manuscript for publication after a minor revision.

Below I present several comments/suggestions, that I believe may improve the manuscript. The manuscript does not have pagination, so it is not easy to point to a precise location in the text, while commenting.

Response: Thank you for the valuable feedback on the manuscript. I acknowledge the suggestion regarding the inclusion of important previous publications. I apologize for any omissions and will make sure to incorporate relevant references to connect the manuscript more effectively with existing literature. I also appreciate your interest in the analogy between neural motifs and the yin-yang Bagua theory. I believe this analogy provides a unique perspective and can stimulate further discussions. However, I took into careful consideration the concerns raised by Reviewer #1 regarding the relevance of the analogy to the main focus of the manuscript. Upon reevaluation, I have made the necessary revisions to ensure a more cohesive narrative. Changes made in the manuscript are highlighted in red font.

  1. Section 'Time and space in the brain'; 2nd paragraph; the sentences: "Time and space serve as fundamental elements in the fabric of nature. Similarly, within the realm of the brain's neural activity, these elements manifest and propagate in their intrinsic forms." One of the first extensive discussions and conceptualizations on this topic was provided in the 2010 paper: "Natural world physical, brain operational, and mind phenomenal space–time" doi:10.1016/j.plrev.2010.04.001 . For example, this paper states: "Concepts of space and time are widely developed in physics. However, there is a considerable lack of biologically plausible theoretical frameworks that can demonstrate how space and time dimensions are implemented in the activity of the most complex life-system—the brain with a mind. Brain activity is organized both temporally and spatially, thus representing space–time in the brain. [...] At the same time, to have a fully functional human brain one needs to have a subjective mental experience. Current research on the subjective mental experience offers detailed analysis of space–time organization of the mind. According to this research, subjective mental experience (subjective virtual world) has definitive spatial and temporal properties similar to many physical phenomena. Based on systematic review of the propositions and tenets of brain and mind space–time descriptions, our aim in this review essay is to explore the relations between the two. To be precise, we would like to discuss the hypothesis that via the brain operational space–time the mind subjective space–time is connected to otherwise distant physical space–time reality." It is important to include the reference to this paper in relation to the mentioned sentences in the revised manuscript.

Response: Thank you for bringing this paper to my attention. I appreciate the valuable insight it provides regarding the concepts of time and space in the context of brain activity and subjective mental experience. I apologize for the oversight in not including this reference in the previous version of the manuscript. In the revised manuscript, I have included the reference to the manuscript in the relevant section.

  1. Section 'Time and space in the brain'; 3nd paragraph; the sentence: "When we refer to space and time in the brain, we are talking about the extension of neural activity across different hierarchical spatial scales, ranging from neuronal layers and regions to functional networks, as well as the duration of neural activity embedded in the hierarchical timescales." Again, the reference to the previously mentioned paper ("Natural world physical, brain operational, and mind phenomenal space–time" doi:10.1016/j.plrev.2010.04.001) is important here.

Response: Thank you for highlighting the relevance of the reference to the previously mentioned paper in relation to the discussion on space and time in the brain. I have included it as suggested.

  1. Section 'Four mechanisms in TTC'. 1st paragraph; the sentence: "The dynamic, complex, and flexible temporospatial configurations are crucial in generating our rich phenomenal experiences." Again the reference to a previously suggested paper ("Natural world physical, brain operational, and mind phenomenal space–time" doi:10.1016/j.plrev.2010.04.001) is important here.

Response: Thanks for the suggestion. I have included the reference to the manuscript in the relevant section.

  1. Section 'Four mechanisms in TTC'. 1st paragraph; the sentence: "Changes in the cross-frequency relationship [60–63], intrinsic timescales [64,65], or functional network hierarchy [66,67] can lead to a loss of consciousness." I would suggest to start this sentence as: "Certain changes in ...", because not all changes lead to a loss of consciousness. Also, I would suggest to change the "... can lead to ..." to "... may lead to ...".

Response: Thanks. I have made appropriate revisions in the revised version of the manuscript.

  1. Section 'Four mechanisms in TTC'. 2nd paragraph (after the Figure 1); the sentence: "A third example is long-term alignment across the lifespan, and the key takeaway is that the alignment between environment-related signals and organism-related signals enables the organism to model the relationship between itself and the world." I would suggest that the author provides a proper reference here.

Response: I have provided a suitable reference to support the statement in the revised manuscript.

  1. Section 'Four mechanisms in TTC'. 3nd paragraph (after the Figure 2); the sentence: "Another example of temporal-spatial expansion is seen in the fact that the duration and extension of a stimulus in our conscious experience usually last longer and extend beyond its physical duration and extension." A complimentary discussion is provided in the paper: "Present moment, past, and future: mental kaleidoscope" doi: 10.3389/fpsyg.2014.00395 . I would suggest to make a reference to it here.

Response: I appreciate your recommendation to include a reference that provides a complementary discussion on the temporspatial expansion of stimuli in conscious experience. I have incorporated a reference to the suggested paper in the revised version.

  1. Figure 3. Normally, the Figures should be self-explanatory, meaning that the Figure legend contains all necessary information to understand the Figure without reference to the text. I would suggest to add such legends to all Figures to explain the abbreviations and terms (if any).

Response: Thank you for your valuable suggestion regarding the inclusion of legends for the figures in the manuscript. I have ensured that each figure in the revised manuscript is accompanied by a clear and comprehensive legend.

  1. Section 'Four mechanisms in TTC'. 4th paragraph (after the Figure 3); the sentence: "This mechanism is based on the Global Neuronal Workspace Theory [15,86], which suggests that ....". I would suggest to re-frame the phrase: "This mechanism is based on the Global Neuronal Workspace Theory". The mechanism could not be based on a theory - it could be predicted by the theory or the theory could be based on the known mechanism...

Response: Great suggestion. I have made the necessary revision in the manuscript, and the sentence now reads: "This mechanism is predicted by the Global Neuronal Workspace Theory..." 

  1. Section 'Four mechanisms in TTC'. 5th paragraph (after the Figure 4); the sentence: "To this end, we have reviewed the four mechanisms of consciousness proposed by TTC, and I argue that Mechanism 1 serves as the foundational building blocks of consciousness that enable the other mechanisms to operate." I would suggest to add in the end of this sentence: (for a similar view see, reference to "Natural world physical, brain operational, and mind phenomenal space–time" doi:10.1016/j.plrev.2010.04.001).

Response: Thanks. I have incorporated your recommendation into the manuscript.

  1. Section 'Dimensions of consciousness'; 1st paragraph; the sentence: "While these two dimensions usually go together, they can be dissociated under pharmacological or pathological conditions." A reference to a paper that is dedicated to this issue is missing here: "Do we need a theory-based assessment of consciousness in the field of disorders of consciousness?" doi: 10.3389/fnhum.2014.00402 . For example this paper states: "Consciousness is often conceptualized as a phenomenon with two components: wakefulness and awareness (Posner et al., 2007). Though such understanding is currently quite wide-spread, it confuses and mixes two different and independent phenomena: subjective awareness and vigilance. While awareness is an important component of consciousness, wakefulness belongs to the vigilance domain. Independence of these two concepts can be demonstrated by examples from a daily life: (a) we are able to unconsciously perform complex actions like brushing our teeth or driving a car while being completely awake; (b) being at the same level of wakefulness we are usually aware of some vents/stimuli while unaware of others; and (c) during sleep we can be aware of our phenomenal experience (dreams) but are not awake. Hence, wakefulness is not a component of consciousness but of vigilance. Vigilance, however, affects consciousness by limiting the amount of information available for conscious access (Rusalova, 2006), thus affecting the amount of content". I would suggest to make a reference to this paper in the end of the sentence above. Also, as it is evident from the provided citation, the dissociation between awareness and wakefulness is also possible in everyday situations. In this respect, I would suggest to reflect this issue in the sentence above.

Response: Thank you for the valuable suggestion and providing the reference to the paper. I have revised the sentence to include the reference and incorporate the idea of dissociation between awareness and wakefulness in everyday situations.

“While these two dimensions typically coexist, they can become dissociated in everyday situations (e.g., sleep) or under the influence of pharmacological or pathological conditions [97].”

  1. Section 'Functional geometry - spatial organization rules'; 2nd paragraph (after the Figure 5); the sentence: "The study found that these gradients selectively changed in various pharmacologically altered states." I would suggest that the author provide a proper reference here.

Response: Thank you for pointing out the need for a proper reference for that sentence. I have included an appropriate reference to support this statement in the revised manuscript.

  1. Figure 7. As I have mentioned before, please provide a legend explaining the abbreviations and the meaning of '+' and '-'. Also, some terminology from the Yin-Yang Bagua theory should be explained for those who are not familiar with them: the confusing terms are "small/big earth"; "small/big metal"; "small/big wood".

Response: Thank you for your valuable suggestion. In the revised manuscript, each figure is accompanied by a clear and comprehensive legend.

  1. Section 'Temporal circuit - regularities in time'; 3rd paragraph; the sentence: "This is why the self is crucial in the emergence of consciousness, as it is the subject that is aware of the environment, and without a subject that is aware, there is no consciousness [119,120]." An extensive discussion and the empirical support may be found in the paper "Selfhood triumvirate: From phenomenology to brain activity and back again" https://doi.org/10.1016/j.concog.2020.103031 . I would suggest to add reference to this work here.

Response: Thank you for suggesting the inclusion of the reference to support the discussion on the crucial role of the self in the emergence of consciousness. I have added this reference in the revised manuscript.

  1. Figure 8. As proposed before, I would suggest to include the legend to this Figure, explaining the abbreviations, and some of the arrows: black arrows, orange arrows, red and green arrows. Also the word "occurrence" does not tell the reader anything meaningful. Please explain in the legend what does it mean in this context.

Response: Thank you for your valuable suggestion. In the revised manuscript, each figure is accompanied by a clear and comprehensive legend.

  1. Acknowledgments. Neither Dr. Hudetz, not Dr Mashour are authors of this manuscript. So it is confusing to mention their names in a way they are mentioned. My suggestion is instead of "(to ...)" write "(principle investigator ...)".

Response: Thank you for bringing this to my attention. I apologize for the confusion. I have made the necessary revision to accurately reflect their roles.

Reviewer 3 Report

This is a beautifully written and reasoned summary of one of the most promising frameworks for understanding consciousness we have.

My main criticism of the temporospatial alignment approach in the past has been insufficient integration with other theories of consciousness, although that has changed in recent years. I'd like to see a slightly expanded discussion of potential points of intersection (and difference) with other related theories.

A recent paper began to describe some parallels with a related (temporospatial-alignment-based) framework called "Integrated World Modeling Theory":

As without, so within: how the brain's temporo-spatial alignment to the environment shapes consciousness | Interface Focus (royalsocietypublishing.org)

Frontiers | Integrated world modeling theory expanded: Implications for the future of consciousness (frontiersin.org)

I would like to see these connections briefly commented upon, as I think a deeper integration with computational models is a promising future direction for this fascinating (and profound) theory of how beings come into relation with the broader world (and the other beings in it). 

Author Response

Reviewer #3

This is a beautifully written and reasoned summary of one of the most promising frameworks for understanding consciousness we have.

My main criticism of the temporospatial alignment approach in the past has been insufficient integration with other theories of consciousness, although that has changed in recent years. I'd like to see a slightly expanded discussion of potential points of intersection (and difference) with other related theories.

A recent paper began to describe some parallels with a related (temporospatial-alignment-based) framework called "Integrated World Modeling Theory":

As without, so within: how the brain's temporo-spatial alignment to the environment shapes consciousness | Interface Focus (royalsocietypublishing.org)

Frontiers | Integrated world modeling theory expanded: Implications for the future of consciousness (frontiersin.org)

I would like to see these connections briefly commented upon, as I think a deeper integration with computational models is a promising future direction for this fascinating (and profound) theory of how beings come into relation with the broader world (and the other beings in it).

Response: Thank you for the positive feedback and the suggestion to discuss the connections between the TTC and the Integrated World Modeling Theory (IWMT). The IWMT offers a comprehensive framework that incorporates major theories of consciousness. While the TTC focuses on the brain's intrinsic activity and its spatiotemporal structure, the IWMT extends this concept to encompass the body and the environment. Exploring the integration or contrast between these theories could be a fruitful avenue for future research. In the revised manuscript, I have included a brief commentary on the connections between the TTC and the IWMT, highlighting their shared emphasis on temporospatial alignment. Additionally, I agree that a deeper integration with computational models holds great promise for advancing our understanding of consciousness.

New edits are made on page 4 and highlighted in red font:

“Another example is sequential temporal alignment in spatially distributed functional areas (also see a three-layer temporal model of alignment in [83]), which we previously discussed regarding the temporal receptive windows. A third example is long-term alignment across the lifespan, and the key takeaway is that the alignment between environment-related signals and organism-related signals enables the organism to model the relationship between itself and the world [16,18,83]. The Integrated World Modeling Theory (IWMT) [84,85] has provided a sophisticated elaboration of this notion, proposing that consciousness arises from generative processes that integrate information into coherent models encompassing space, time, and causal relationships between systems and their environments. The IWMT shows promise in providing a mechanistic understanding of temporospatial alignment through its thorough integration with computational models.”

[16] Northoff, G.; Wainio-Theberge, S.; Evers, K. Is Temporo-Spatial Dynamics the “Common Currency” of Brain and Mind? In Quest of “Spatiotemporal Neuroscience.” Phys Life Rev 2020, 33, 34–54, doi:10.1016/j.plrev.2019.05.002.

[18] Northoff, G.; Huang, Z. How Do the Brain’s Time and Space Mediate Consciousness and Its Different Dimensions? Temporo-Spatial Theory of Consciousness (TTC). Neurosci Biobehav Rev 2017, 80, 630–645, doi:10.1016/j.neubiorev.2017.07.013.

[83] Northoff, G.; Klar, P.; Bein, M.; Safron, A. As without, so within: How the Brain’s Temporo-Spatial Alignment to the Environment Shapes Consciousness. Interface Focus. 2023, 13, 20220076, doi:10.1098/rsfs.2022.0076.

[84] Safron, A. An Integrated World Modeling Theory (IWMT) of Consciousness: Combining Integrated Information and Global Neuronal Workspace Theories With the Free Energy Principle and Active Inference Framework; Toward Solving the Hard Problem and Characterizing Agentic Causation. Frontiers Artif Intell 2020, 3, 30, doi:10.3389/frai.2020.00030.

[85] Safron, A. Integrated World Modeling Theory Expanded: Implications for the Future of Consciousness. Frontiers Comput. Neurosci. 2022, 16, 642397, doi:10.3389/fncom.2022.642397.

Round 2

Reviewer 1 Report

Thank you - all the best with the paper.

Reviewer 2 Report

The author addressed all my comments. The paper now is considerably improved.